# The weekly P25 of the age of the influenza-like illness shows a higher correlation with COVID-19 mortality than rapid tests and could predict the evolution of COVID-19 pandemics in sentinel surveillance, Piura, Perú, 2021

**Víctor Raúl Ocaña Gutiérrez**[1☯¤‡]*, **Rodolfo Arturo González Ramírez**[1,2☯‡], **Víctor Alexander Ocaña Aguilar**[1,2☯], **Nadia Gabriela Ocaña Aguilar**[2☯], **Carlos Enrique Holguín Mauricci**[3☯]

**1** School of Medicine, Cesar Vallejo University, Piura, Perú, **2** Department of Public Health, School of Medicine, National University of Cajamarca, Cajamarca, Perú, **3** Department of Public Health, School of Medicine, National University of Piura, Piura, Perú

☯ These authors contributed equally to this work.
¤ Current address: Infectious Diseases Area, Health Center I, Piura, Perú
‡ VROG and RAGR are Joint Senior Authors
* vocanag01@gmail.com, vocanag@unp.edu.pe

## Abstract

### Goal

To describe the dynamics of syndromic surveillance of ILI cases in seasonal and COVID-19 pandemic scenarios.

### Methodology

A descriptive study of the epidemiological behavior of ILI in the seasonal and COVID-19 pandemic scenarios. Of a sample of 16,231 cases of ILI from 2013 to 2021, the features of cases from 68 weeks before and during the pandemic were selected and compared; weekly endemic channels were built; data fluctuations on the trend of ILI cases were analyzed; and estimated weekly correlations between weekly P25 age, cases confirmed by rapid tests, and mortality from COVID-19. To analyze clinical-epidemiological and mortality data, Student's t test, Mann-Whitney U, Chi2, Spearman's Ro, polynomial, and multinomial regression with a 95% confidence interval were used.

### Results

During the COVID-19 pandemic, those most affected with ILI were: adults and the elderly; higher median age; autochthonous cases predominated; a lower proportion of other syndromes; delays in seeking care; and a higher rate of pneumonia attack than in the seasonal period (p< 0.01). Rapid tests (serological and antigenic) confirmed 52.7% as COVID-19. Two ILI pandemic waves were seasonally consistent with confirmed COVID-19 cases and district mortality with robust correlation (p<0.01) before and during the pandemic, especially

**Data Availability Statement:** All COVID-19 Mortality files are available in the database of the Peruvian Ministry of Health MINSA Open Data and Knowledge Management in COVID-19, Perú: https://www.minsa.gob.pe/datosabiertos/. Through this link, it is possible to access COVID-19 mortality data in Piura district and COVID-19 cases reported from I-4 Pachitea Health Center, Piura, Peru, and the ILI database has been sent in an Excel.xls file because ETI surveillance data base is not accessible online, it is only accessible through either the author at email: vocanag01@gmail.com or as a Supporting Information file here.

**Funding:** The authors received no specific funding for this work.

**Competing interests:** This does not alter our adherence to the PLOS ONE policy on sharing data and materials.

the ILI weekly P25 age, which has a more robust correlation with mortality than ILI and rapid tests (p<0.01) whose endemic channels describe and could predict the evolution of the pandemic (p<0.01).

## Conclusions

The pandemic changed the clinical and epidemiological behavior of ILI, and the weekly P25 of age is a more robust indicator to monitor the COVID-19 pandemic than a rapid test and could predict its evolution.

## Introduction

In December 2019, the first outbreak of SARS-Cov-2 or COVID-19 began in Wuhan, China [1,2]. On January 30, 2020, the pandemic situation was declared, and on January 31, it was recommended that all countries be prepared to face its expansion [3]. In Peru, the first cases were detected in week 10 of the year 2020 in the Lima region; since March 6 and in the same week in the Piura Region in northern Peru [4], the pandemic has been spreading and taking a fragmented and weakened health system by surprise, facilitating its rapid spread [5]. Control measures were implemented in the country, with containment, mitigation, and suppression approaches and a COVID-19 surveillance system based on laboratory diagnosis with molecular, immunological, and mainly rapid antigen tests [6,7]. However, due to the limited availability of these tests, an underreporting of COVID-19 cases was generated in the country, as has been demonstrated. So the official data did not show the absolute magnitude of the pandemic [8,9].

Syndromic sentinel surveillance, recommended by the WHO [10,11], is a strategy used by the CDC in the US and other places in the world for chronic and infectious diseases that links the clinic with the laboratory [12–14], which has been implemented at the Pachitea Health Center since 2013 in the city of Piura, in northern Peru, where influenza-like illness (ILI) is monitored. ILI syndromic surveillance based on clinical diagnosis of acute respiratory diseases includes confirmed and unconfirmed COVID-19 cases with laboratory tests. Its use during the pandemic could provide better information regarding morbidity and mortality related to COVID-19 if we consider the low sensitivity and specificity of rapid tests, in which the average sensitivity is usually 56.2% (95% CI: 29.5 to 70.8%) [15], since the WHO recommends 'acceptable' sensitivity $\geq$ 80% and specificity $\geq$ 97%, and could be as similar to RT-PCR [15]; For this reason, it is necessary to identify clinical and epidemiological criteria to follow the weekly evolution of the pandemic and validate its representativeness by correlating it with the weekly evolution of mortality, taking into account evidence of a strong correlation of mortality with COVID-19 cases at the beginning of the pandemic [16–18], considering mortality as a gold standard for surveillance of the pandemic in this first level of care.

The present study compared the characteristics of ILI cases in seasonal and pandemic scenarios. The dynamic characteristics of the weekly ILI syndromic surveillance curve and its correlation with the weekly curve of confirmed cases with rapid tests and mortality from COVID-19 during the pandemic were also described. There are mathematical models that try to establish forecasts of the pandemic full of inaccuracies due to the underreporting of cases [8,19]; therefore, it is necessary to find other, perhaps more accurate, pandemic monitoring indicators. Thus, the aim is to evaluate the usefulness and scope of syndromic sentinel surveillance of ILI in the COVID-19 pandemic.

## Materials and methods

In the district of Piura, located on the northern coast of Peru (584,000 inhabitants), there is a health care network with hospitals and first-level health centers. One of the first-level health-care facilities is the Pachitea Health Center, located in the center of the city, with 50,800 inhabitants in its jurisdiction. Since 2013, a surveillance protocol for syndromic ILIs has been applied in outpatient care. In ILI sentinel surveillance, an ILI case has been operationally defined as a person of any age who comes to the health facility with fever plus one or all (cough, sore throat, runny nose, or lower respiratory system symptoms) and with less than 15 days of illness. To avoid duplication of records for the same disease, as of 2013, it was established that only incident cases would be systematically recorded in this surveillance, and then all cases that returned again before 15 days of illness onset were excluded. A case of COVID-19 was confirmed through immunologic and rapid antigen tests, and serological or molecular tests did not apply. Using molecular, antigenic, and serological tests, as well as epidemiological links, deaths from COVID-19 were confirmed in the district of Piura.

From the sentinel surveillance database, which includes those who attended the health facility between 2013 and 2021, a census sample of 16,231 registered ILI cases was obtained (Fig 1). Then, data on age, sex, origin, duration of illness, and pneumonia were obtained from this sample of ILI cases. Other concomitant febrile syndromes were also recorded, such as abdominal, genitourinary, undifferentiated, eruptive, neurological, arthritic, and hemorrhagic syndromes. The clinical-epidemiological data of 5,998 ILI patients registered in sentinel surveillance from weeks 10 of 2018 to 26 of 2019 (seasonal scenario) were compared with those from weeks 10 of 2020 to 26 of 2021 (pandemic scenario), 68 weeks of each period. ILI cases were distributed by epidemiological week, and endemic channels were built by epidemiological weeks based on analysis of measures of the central tendency of p25 weekly age cases (25, 50, and 75 percentile) between the years 2013 and 2020 week 10 (before the COVID-19 pandemics) using Excel for Windows. These endemic ILI channels were used to monitor the

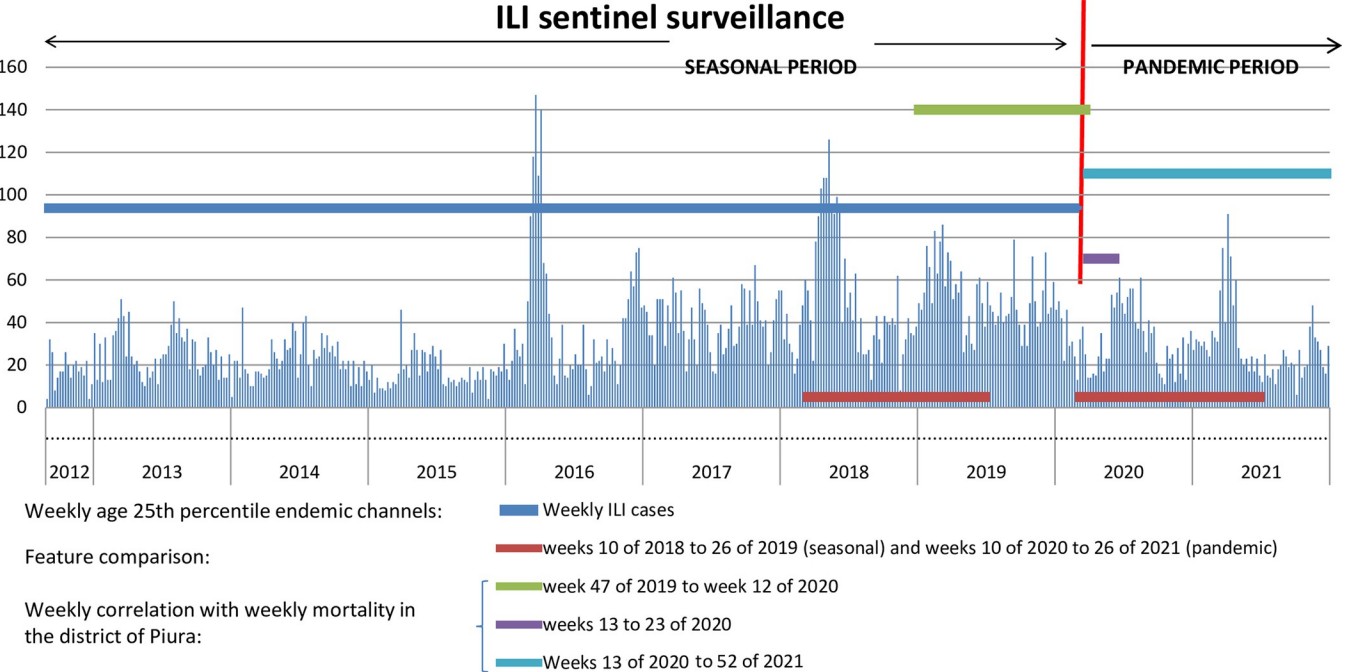

**Fig 1. ILI cases per week.** By time periods, several samples were selected for comparative, correlational, and predictive analysis.

behavior of the weekly p25 of age during the COVID-19 pandemic (from week 10 of 2020 to week 52 of 2021). Weekly frequency data for the 1419 cases of COVID-19 confirmed by rapid antigen testing at the establishment were also analyzed. A publicly accessible database on general and COVID-19 mortality, recorded in the district of Piura from 2019 to 2021, was also used. In addition, the weekly medians of age, P25, and P50 corrected by Tukey hinges of the age of ILI cases by epidemiological weeks from the years 2013 to 2021 were also analyzed. For the time series from 2013 to 2021 of medians, P25 and P75 of the weekly age of ILI cases were analyzed using the EPIPOI [20] software with the polynomial data detrending function.

The data were analyzed using the software R Studio version 4.2.2 with the R command module. For quantitative variables, using the Kolmogorov-Smirnov Normality test for a weekly count of variables, Spearman correlations were estimated on the weekly behavior curves in the sentinel center between 2,299 ILI cases (weekly median age and 25th percentile), 1,419 COVID-19 confirmed cases, and 2,218 COVID-19 deaths from the Piura district. Hypothesis tests were also used for trend measures central, T-score for the mean of independent samples, and Mann-Whitney U for comparison of medians. For qualitative variables, non-parametric Chi 2 tests and their corrections were used, as well as for quantitative variables, Spearman's Rho estimation for correlation analysis and polynomial regression for prediction of weekly curves. The hypotheses were contrasted with a 95% confidence interval.

Since this study was carried out using secondary data from daily records of patients presenting with ILI symptoms and mortality from COVID-19, the data collected from these sources was recorded in a format protecting the identifiable data of the subjects with encrypted coding (S1 Dataset, S1 and S2 Files). Mortality data have been obtained from a public access source (S3 File). For these reasons, the research ethics committee of Cesar Vallejo University, while this study was approved, waived informed consent. In addition, authorization was obtained from the director responsible for custody of the data for access in the health establishment; for this reason, syndromic sentinel surveillance data are not available online.

## Results

### ILI in the seasonal and COVID-19 pandemic periods

When comparing the characteristics of ILI cases from the seasonal and pandemic periods, the same proportion of ILI cases by sex was found in both periods (P > 0.05). According to age, in the seasonal period, the frequency of cases increased as age decreased, observing a broad-based pyramidal structure in both sexes, while during the pandemic, it was more frequent in the adult population and decreased towards childhood or old age. During the seasonal period, the 25th percentile value of the weekly age was three years, while during the COVID-19 pandemic, it reached 22 years; there was also an increase in the median and the 75th percentile. By age groups, cases in children and adolescents were more frequent in the seasonal periods, while in the pandemic period, cases in adults and older adults were more frequent. The population went to the health center in a greater proportion from within the jurisdiction during the pandemic period and during the seasonal period from outside. The seasonal period showed a higher proportion of cases that simultaneously presented ILI and other syndromes than in the pandemic. During the pandemic, the population went to the health center later than in the seasonal period. Pneumonia attack rates in ILI cases during the COVID-19 pandemic were significantly higher than in the seasonal period; in adults, it is significant, unlike the other ages (Table 1). Of 2,692 cases reported as suspected COVID-19 during the pandemic period, 1,419 (52.7%) were laboratory confirmed (rapid tests such as 46% immunology and 54% antigen) as COVID-19 cases, and it was found that 43.4% of the reported negatives and 40.1% of those confirmed as COVID-19 were ILI cases (P > 0.05).

**Table 1. Characteristics of ILI cases in seasonal and pandemic periods of COVID-19, Pachitea Health Center.**

| Features | PERIOD | | | | ESTADISTIC | P valor |
|---|---|---|---|---|---|---|
| | Seasonal (*) | | COVID-19 Pandemic (**) | | | |
| | N = 3689 | % | N = 2299 | % | | |
| **SEX** | | | | | | |
| Male | 1768 | 47.9 | 1112 | 48.4 | Chi$^2$ Pearson = 0.11 | 0.74 |
| Female | 1921 | 52.1 | 1187 | 51.6 | | |
| **Age** | | | | | | |
| Minimum | 0.01 | | 0.01 | | U de Mann-Whitney = 2324677.0 | |
| Maximum | 100 | | 95 | | | |
| Rank | 99.99 | | 94.99 | | | |
| 25th percentile (***) | 3 | | 22 | | | |
| Median | 10 | | 35 | | | **<0.01** |
| 75th percentile (***) | 30 | | 50 | | | |
| **Jurisdiction of cases** | | | | | | |
| Autochthonous | 2194 | 59.5 | 1444 | 62.8 | Chi$^2$ Pearson = 6.61 | **0.01** |
| Imported | 1495 | 40.5 | 855 | 37.2 | | |
| **ILI and other febrile syndromes (****)** | | | | | | |
| ILI | 3289 | 89.2 | 2274 | 94.6 | Chi$^2$ R. Verisimilitude = 57.54 | **<0.01** |
| ILI and ABD | 110 | 3 | 41 | 1.8 | | |
| ILI and GUR | 106 | 2.9 | 40 | 1.7 | | |
| ILI and IND | 81 | 2.2 | 24 | 1 | | |
| ILI and ERU | 44 | 1.2 | 8 | 0.3 | | |
| ILI and others | 59 | 1.6 | 12 | 5 | | |
| **ILI and other manifestations** | | | | | | |
| ILI and otitis | 90 | 2.4 | 7 | 0.3 | Chi$^2$ R. Verisimilitude = 103.68 | **<0.01** |
| ILI and oral ulcers | 32 | 0.9 | 2 | 0.1 | | |
| ILI and conjunctivitis | 19 | 0.5 | 6 | 0.3 | | |
| **Pneumonia** | | | | | | |
| Age group | Pneumonia/ILI cases | | Pneumonia/ILI cases | | CMLE rate ratio | Mid-p exact |
| Kids | 38/1916 | 1.98 | 11/315 | 3.49 | 1.761 | 0.11 |
| Adolescent | 0/246 | 0 | 0/101 | 0 | – | – |
| Young boys | 9/590 | 1.53 | 10/526 | 1.9 | 1.246 | 0.6386 |
| Adults | 15/654 | 2.29 | 61/1056 | 5.78 | 2.52 | **<0.01** |
| Older adults | 35/283 | 12.37 | 46/301 | 15.28 | 1.236 | 0.3476 |
| **Total** | **97/3689** | **2.63** | **128/2229** | **5.57** | **2.117** | **<0.01** |
| **Sick time attending$^{\&}$** | | | | | | |
| Mean (days) | 1.97 (DE± 1.84) | | 2.35 (DE± 2.28) | | t = 6.78 | **<0.01** |

(*) Seasonal period (week 10 of the year 2018 to 26 of the year 2019, 68 weeks).

(**) COVID-19 pandemic period (weeks 10 of the year 2020 to 26 of the year 2021, 68 weeks).

(***) Tukey hinges.

(****) GUR = Genitourinary; IND = Undifferentiated; ABD = Abdominal; ERU = Eruptive.

$^{\&}$ In both periods, a minimum of 0 days and a maximum of 15 days are allowed.

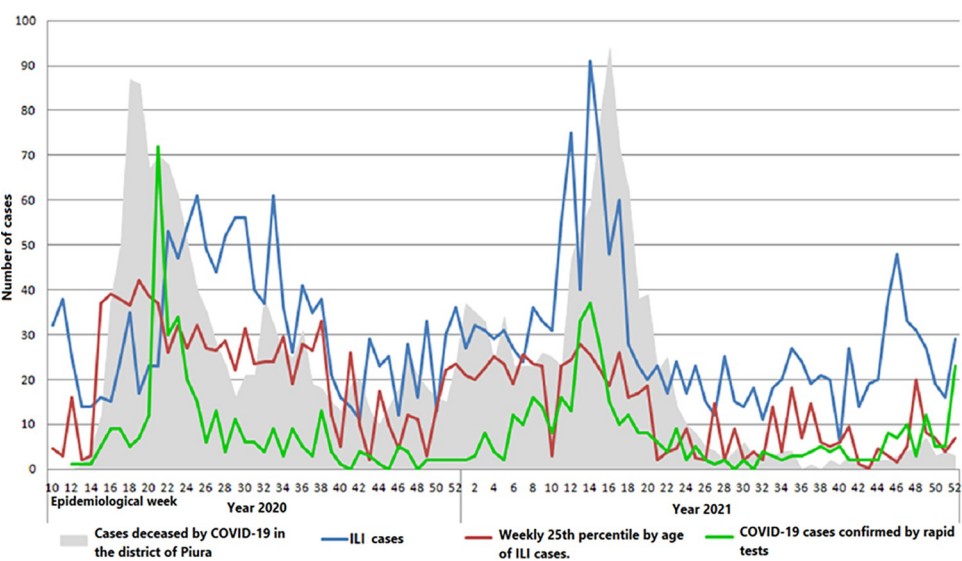

**Fig 2. Weekly comparison between ILI cases, COVID-19 cases by rapid tests, and the 25th percentile of age of ILI cases with mortality cases by COVID-19 in the district of Piura.**

### Temporal analysis and correlations

In the seasonal period (week 47 of the year 2019 and week 12 of the year 2020), a significant and robust correlation was found between ILI cases per week registered in the sentinel establishment and the weekly general mortality from all causes in the Piura district (Spearman´s rho = 0.718, p < 0.01).

During the COVID-19 pandemic period, similarity was observed in the trends of weekly ILI cases, the median value, the 25th percentile of age, and the number of COVID-19 cases confirmed by rapid tests in the sentinel establishment with weekly mortality by COVID-19 in the district of Piura. (Fig 2). In the first wave of the COVID-19 pandemic, around week 30 of the year 2020, there was overall excess mortality from all causes about four times higher than usual, and throughout the pandemic, it always remained highest over expected mortality. In the district of Piura, almost coincidentally in the same period, the two weekly pandemic waves of mortality linked to COVID-19 occurred between weeks 14 and 28 of 2020 and 2021, with the maximum peaks of cases in weeks 18 and 16, respectively (Fig 2). At the beginning of the pandemic period in the first wave of COVID-19 between weeks 13 and 23 of 2020, in which there was a sustained increase in cases, a strong correlation was found between ILI cases per week and overall weekly mortality from all the causes (Spearman's rho = 0.712, p < 0.05).

During the pandemic period between week 13 of 2020 and week 52 of 2021, when estimating the correlation between the specific weekly Mortality due to COVID-19 in the district of Piura and the weekly cases of ILI in the sentinel establishment, there is a slight to moderate correlation (Spearman's rho = 0.463, p<0.01); with the median weekly age of ILI cases in the sentinel establishment, there is a moderate to robust correlation (Spearman's rho = 0.686, p < 0.01); with the 25th percentile of weekly age of ILI cases of the sentinel establishment there is a robust correlation (Spearman's rho = 0.716, p < 0.01); also with the weekly cases of COVID-19 of the sentinel establishment confirmed by rapid tests, there is a moderate correlation (Spearman's rho = 0.587, p < 0.01). There is also a moderate correlation between weekly COVID-19 cases confirmed by rapid tests and weekly ILI cases (Spearman's rho = 0.521, p < 0.01), and between weekly COVID-19 cases ruled out by rapid tests and weekly ILI cases, there is a robust correlation (Spearman's rho = 0.770, p < 0.01). (Fig 2).

## Evolution of the 25th percentile of the weekly age of ILI cases in the seasonal period and the COVID-19 pandemic

Since the most robust correlation was between the 25th percentile of the weekly age of ILI cases and the weekly mortality caused by COVID-19, its epidemiological behavior was described during seasonal and pandemic periods. Thus, in the seasonal period, the weekly epidemiological behavior of the 25th percentile of the weekly age cases with ILI from the year 2013 until week 52 to 2021 was analyzed for seasonality, anomalies above the 95% CI, and grids of the 25th percentile of the weekly age with the polynomial data detrending function. Thus, in the seasonal period, it was observed that the P25 weekly age of ILI always remained at values close to 5 years, with some peaks of epidemic waves; these waves lasted a few weeks in the years 2014, 2015, 2017, and 2018. During the COVID-19 pandemic period from the year 2020 (after week 10) and all of 2021, the 25th percentile of weekly age clearly plots the two COVID-19 pandemic waves of the year 2020 and 2021 that present anomalies beyond 1.96 SD, which are of greater amplitude and duration than the epidemic waves of the pre-pandemic period. Likewise, the 25th percentile values of weekly age during the pandemic period remained well above the seasonal one (Fig 3). Also, when it was constructing endemic channels (using medians and P25 and P75 for weekly age) with the value of the weekly 25th percentile of the age of ILI cases between week 1 of the year 2013 to week 10 of 2020 (from the

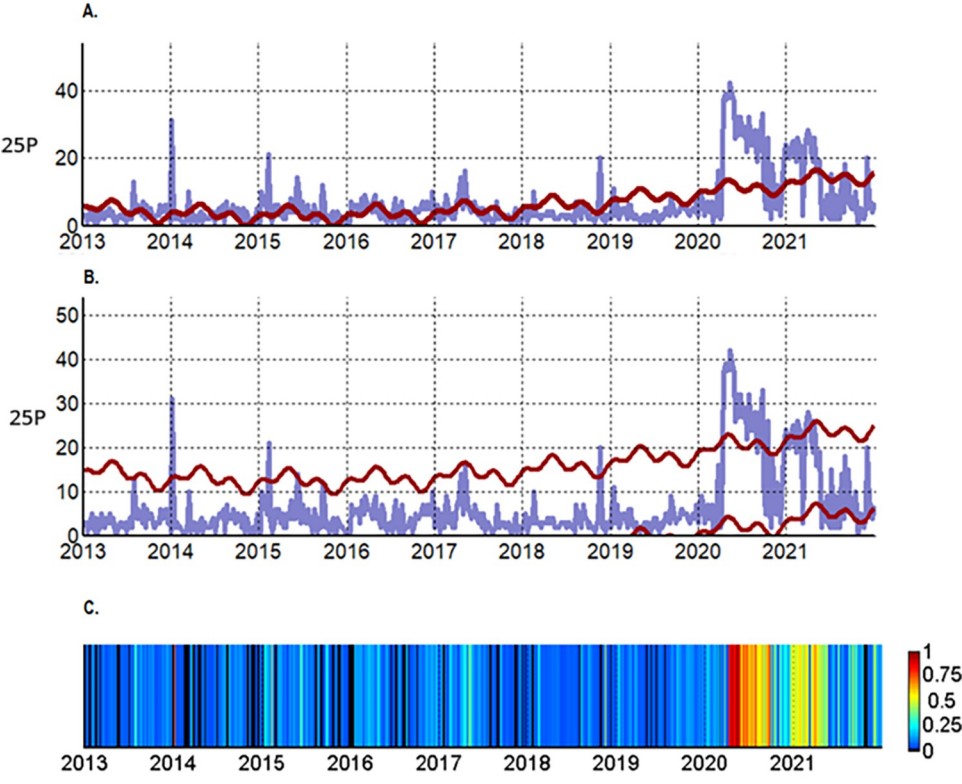

A.  Seasonality of weekly age 25P in patients with ILI

B.  Anomalies beyond 1.96 stand. dev. of weekly age 25P in patients with ILI

C.  Grids of weekly age 25P in patients with ILI

**Fig 3. Seasonality, anomalies above the 95% CI, and grids of the 25th percentile of weekly age.**

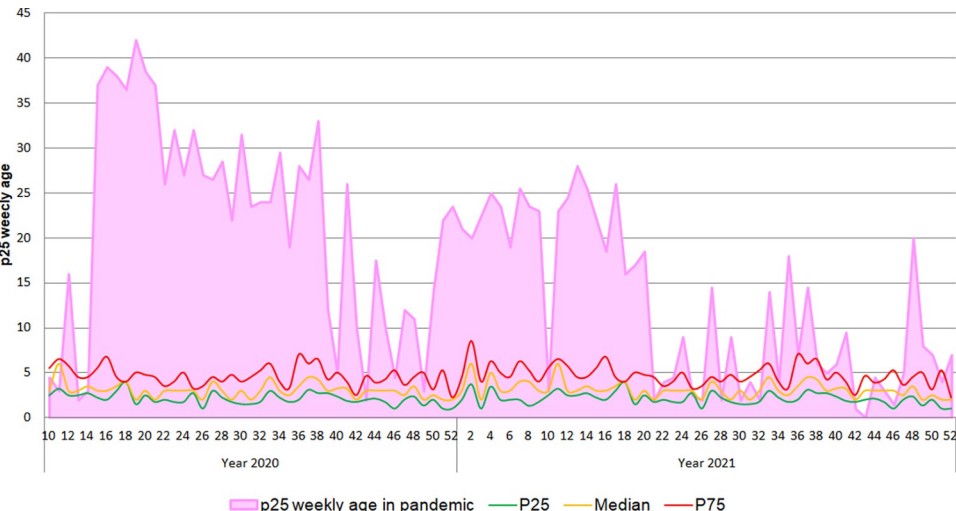

**Fig 4. Weekly epidemiological behavior of the 25th percentile of the age of weekly ILI cases in the COVID-19 pandemic period.**

seasonal period), throughout the 92 weeks of the pandemic period, when these channels were used to evaluate the behavior of the 25th percentile of weekly age, they clearly showed that there was an increase in their values at the epidemic level throughout the period, plotting the two pandemic waves that were similar to COVID-19 mortality curves (Figs 2–4). Likewise, throughout the 92 weeks of the pandemic period, a trend towards a decrease in the value of the 25th percentile of weekly age was observed from the first wave (week 10 of 2020) to week 52 of 2021 (Fig 4). Then, when applying a polynomial regression of degree 2, a significant forecast estimate of a decreasing trend of the pandemic was found (multiple R-squared: 0.351, adjusted R-squared: 0.337, F-statistic: 24.84 on 2 and 92 DF, p < 0.01). In this way, the model significantly predicts a 33.7% tendency to return to pre-pandemic levels of the 25[th] percentile for weekly age value (Fig 5). Using multimodal logistic regression, this prediction is only affected by the age of the cases (p < 0.01), while sex, origin, and time of ILI disease that attend care do not influence each other independently or in combination (p > 0.05).

## Discussion

ILI syndromic sentinel surveillance has been used since before the COVID-19 pandemic [21]. In this sense, regarding the weekly evolution of the effects of the pandemic on the frequency of ILI cases, this study demonstrates that sentinel syndromic surveillance can monitor the weekly evolution of the pandemic in a more representative way than doing so with COVID-19 cases (confirmed by rapid tests) and could significantly predict its evolution. Differences in the frequency of presentation of some characteristics of ILI between the seasonal and pandemic periods were observed, and almost half of the COVID-19 cases in ILI patients using rapid COVID-19 tests were detected [22]. In this way, the introduction of the Sars-Cov-2 virus into the population caused changes in the pattern of clinical and epidemiological behavior of ILI. It is interesting to find that the two pandemic waves of ILI cases detected and linked to the circulation of the SARS-Cov 2 virus and the increase in mortality occurred simultaneously in the same period of time between the end of summer and the beginning of autumn; this indicates a seasonal trend that coincides with seasonal outbreaks of influenza viruses in this region [23].

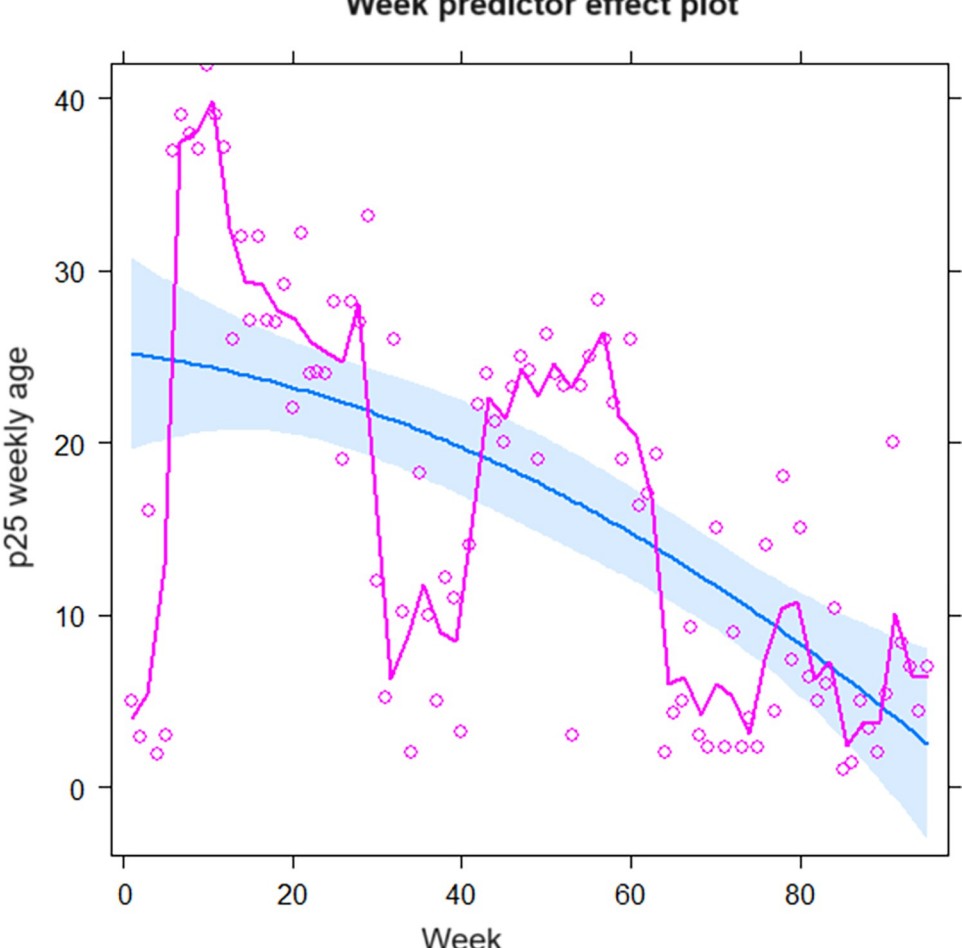

**Fig 5. Estimation of the prediction effect of weekly p25 age on the evolution of the COVID-19 pandemic period** (magenta lines and circles represent p25 weekly age, the blue line is a polynomial regression of order 2, and the light blue shading is a 95% confidence interval).

Despite the fact that the sentinel facility cares for 50,800 of the 584,000 inhabitants of the district of Piura, in the sentinel establishment there is a significant magnitude of weekly correlation between COVID-19 mortality from the district of Piura with cases of ILI, median, and 25th percentile of age; however, to a lesser extent, with COVID-19 cases confirmed by rapid tests. This shows the advantages and representativeness of sentinel surveillance of syndromic ILI during the pandemic compared to what was done by rapid testing, especially the 25th percentile of weekly age of ILI cases, which had the strongest correlation with district mortality. It is probable that the 25th percentile of the weekly age in the pandemic period graphs with greater precision the two pandemic waves in the years 2020 to 2021 than rapid tests. This was corroborated in a study at the beginning of the pandemic, in the first pandemic wave, in which there was a robust weekly correlation between all-cause mortality and ILI cases; too, there was great sensitivity to detect outbreaks due to new respiratory viruses [24]. The robust correlation between the weekly median and weekly 25th percentile of the age of ILI cases and mortality from COVID-19 is explained because the number of receptors for the SARS-Cov-2 virus decreases from the elderly to children, so children are less likely to become infected [25], and also because minors have an immature and developing immune system, as well as environmental [25] and host factors such as polymorphisms and mutations of the ACE2

(Angiotensin-converting enzyme type 2) and TMPRSS2 genes (transmembrane protein genes having serine protease activity) [26]; therefore, older people are the most prone to and affected [24]. In the seasonal period, the weekly median age had low values, close to 10 years, while during the COVID-19 pandemic period, it rose to 35 years. In this regard, it is known that seasonal coronaviruses also present these characteristics with a median age of 33 years, so they did not contribute significantly to increasing the median age in the seasonal period due to their low average prevalence of 4% [27].

The higher proportion of ILI cases without other febrile syndromes in the pandemic period compared to the seasonal one is similar to what was reported in Italy at the beginning of the pandemic, when COVID-19 emerged in the midst of a wave of Influenza H1N1 and RSV viruses and displaced influenza, and RSV was the only circulating virus [28], although the mitigation measures that were implemented also contribute to this, as was found in an outbreak of Influenza A H3N2 in the Kingdom of Cambodia [29] and in Shanghai, where the impact of non-pharmacological measures to contain COVID-19 is measured [30]. This same phenomenon also occurred with the spread of Novel Influenza A H1N1 (pH1N1) in Perú [31].

ILI is claimed to induce misclassification with COVID-19 cases [32]; however, in this study, the robust correlation of the 25th percentile of weekly age of ILI cases with COVID-19 mortality, which is better than with laboratory-confirmed COVID-19 cases, demonstrates the opposite; this is similar to that was reported in China, where an abnormal increase in ILI is anticipated one month before reports of COVID-19 atypical pneumonia, while in the routine monitoring system, the peak of ILI occurred 20 days before the declaration of the massive official alert about the epidemic [33], also in France [34], in the USA [35], another in China [18], and in Germany on excess mortality [17].

The similarity by sex in the involvement of ILI in seasonal and pandemic periods indicates that initially the spread of COVID-19 is similar in both sexes because people go to the health center with an average of 2.35 days of illness, although the complications are more frequent in adult males later in the course of the disease [36]. The strengths of the study are the size of the census sample, the weekly data available from sentinel surveillance, and mortality since 2013. Some of the weaknesses of the study are that during the start of the pandemic, coinciding with the increase in ILI, the first wave (weekly mortality has lately represented it), by order of the health authorities, the sentinel center was not fully operational and was only limited to meeting the demand for cases with warning signs of COVID-19 and timely referral [7], and there were delays in the availability of rapid tests. Thus, the ILI cases in the first pandemic wave do not reflect the reported magnitude of mortality; however, this was overcome with the analysis of the 25th percentile of weekly age, which allows us to reconstruct the first wave of COVID-19. Another limitation of sentinel surveillance is the impossibility of obtaining the reference population to estimate incidence or prevalence rates.

ILI sentinel surveillance contributions in a complementary way are similar to other studies [12,32,34,35]. In this way, it will facilitate the identification of the causal agents of ILI outbreak peaks by linking them with laboratory surveillance. Based on ILI surveillance, the application of Bayesian statistics could estimate the prediction of the end of the pandemic. It could also be used to expand the potential of syndromic surveillance to other settings and to evaluate the impact of health interventions; likewise, this would facilitate informed and timely decision-making in public health interventions.

## Conclusions

Sars-Cov-2 virus´s dissemination into the vulnerable population changed the clinical and epidemiological aspects of ILI. Such as a higher age of affection, complications, and mortality in

comparison to the pre-pandemic period, and so on. There is also a tendency for seasonal outbreaks of ILI spikes linked to COVID-19 to occur. Clinical surveillance for ILI by epidemiological week, mainly weekly 25th-percentile of age, shows a significantly stronger correlation with COVID-19 mortality than found with laboratory surveillance. Therefore, the 25th percentile of weekly age could be an indicator of the COVID-19 pandemic and its future evolution. Therefore, a downward trend in its value from the maximum peak reached at the beginning of the pandemic onwards could predict its end in the future, and according to these projections, it is likely that, when it reaches steadily a value around 3 with seasonal variations, then we will probably return to the pre-pandemic period. In this way, the weekly 25-year-old ILI percentile could help in areas with insufficient laboratory support, generate and maintain local surveillance. Strengthening the response of the first level of care in the opportune detection of viral respiratory outbreaks. It could also serve as an indicator to assess the impact of health interventions, and this could probably have a very low cost compared to rapid tests.

## Supporting information

**S1 Dataset. ILI-2012-2021Dec.**
(XLS)

**S1 File. Correlations-ILI-MedianAge and COVID-19-Mortality.**
(XLSX)

**S2 File. Average Median-Percentils 95IC wAge-2012-2021.**
(XLSX)

**S3 File.**
(DOCX)

## Acknowledgments

To the Cesar Vallejo SAC University of Peru for its support in the research activities of this study, and to the I-4 Pachitea Health Center for providing its facilities for this study and allowing access to the data.

## Author Contributions

**Conceptualization:** Víctor Raúl Ocaña Gutiérrez, Rodolfo Arturo González Ramírez, Nadia Gabriela Ocaña Aguilar, Carlos Enrique Holguín Mauricci.

**Data curation:** Víctor Raúl Ocaña Gutiérrez, Víctor Alexander Ocaña Aguilar.

**Formal analysis:** Víctor Raúl Ocaña Gutiérrez, Víctor Alexander Ocaña Aguilar, Nadia Gabriela Ocaña Aguilar, Carlos Enrique Holguín Mauricci.

**Funding acquisition:** Víctor Raúl Ocaña Gutiérrez.

**Investigation:** Víctor Raúl Ocaña Gutiérrez, Víctor Alexander Ocaña Aguilar, Nadia Gabriela Ocaña Aguilar.

**Methodology:** Víctor Raúl Ocaña Gutiérrez, Rodolfo Arturo González Ramírez, Víctor Alexander Ocaña Aguilar, Nadia Gabriela Ocaña Aguilar.

**Project administration:** Víctor Raúl Ocaña Gutiérrez, Nadia Gabriela Ocaña Aguilar, Carlos Enrique Holguín Mauricci.

**Resources:** Víctor Raúl Ocaña Gutiérrez, Rodolfo Arturo González Ramírez.

**Software:** Víctor Raúl Ocaña Gutiérrez, Víctor Alexander Ocaña Aguilar, Nadia Gabriela Ocaña Aguilar.

**Supervision:** Víctor Raúl Ocaña Gutiérrez, Rodolfo Arturo González Ramírez, Nadia Gabriela Ocaña Aguilar, Carlos Enrique Holguín Mauricci.

**Validation:** Víctor Raúl Ocaña Gutiérrez, Rodolfo Arturo González Ramírez, Carlos Enrique Holguín Mauricci.

**Visualization:** Víctor Raúl Ocaña Gutiérrez.

**Writing – original draft:** Víctor Raúl Ocaña Gutiérrez, Rodolfo Arturo González Ramírez, Víctor Alexander Ocaña Aguilar, Nadia Gabriela Ocaña Aguilar, Carlos Enrique Holguín Mauricci.

**Writing – review & editing:** Víctor Raúl Ocaña Gutiérrez, Rodolfo Arturo González Ramírez, Víctor Alexander Ocaña Aguilar, Nadia Gabriela Ocaña Aguilar, Carlos Enrique Holguín Mauricci.

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
