## [Decision Letter · Decision Letter 0]

16 May 2023

PONE-D-23-01394In sentinel surveillance the weekly P25 of age of the influenza-like illness shows a higher correlation with COVID-19 mortality than rapid tests and could predict the evolution of COVID-19 pandemics, Piura, Peru-2021.PLOS ONE

Dear Dr. OCANA GUTIERREZ,

Thank you for submitting your manuscript to PLOS ONE. After careful consideration, we feel that it has merit but does not fully meet PLOS ONE’s publication criteria as it currently stands. Therefore, we invite you to submit a revised version of the manuscript that addresses the points raised during the review process.

The manuscript has some problems in the description of the methodology and other problems in the language. I request that a comprehensive review be carried out as directed by reviewers 1 and 3.

We are looking forward to a new version.

We look forward to receiving your revised manuscript.

Kind regards,

André Ricardo Ribas Freitas

Academic Editor

PLOS ONE

“NO”

“The opinions expressed in this article belong to the authors and do not necessarily reflect the official policy or position of the Ministry of Health of Peru, Universidad Cesar Vallejo SAC of Peru. Several of the authors are employees of the Peruvian Ministry of Health and/or the Cesar Vallejo SAC University of Peru. This work was prepared as part of his official duties. The authors declare no conflict of interest.”

Additional Editor Comments:

The manuscript has some problems in the description of the methodology and other problems in the language. I request that a comprehensive review be carried out as directed by reviewers 1 and 3.

We are looking forward to a new version.

Reviewers' comments:

Reviewer's Responses to Questions

**Comments to the Author**

1. Is the manuscript technically sound, and do the data support the conclusions?

Reviewer #1: No

Reviewer #2: Yes

Reviewer #3: Partly

2. Has the statistical analysis been performed appropriately and rigorously? 

Reviewer #1: I Don't Know

Reviewer #2: Yes

Reviewer #3: I Don't Know

3. Have the authors made all data underlying the findings in their manuscript fully available?

Reviewer #1: No

Reviewer #2: No

Reviewer #3: No

4. Is the manuscript presented in an intelligible fashion and written in standard English?

Reviewer #1: No

Reviewer #2: Yes

Reviewer #3: No

5. Review Comments to the Author

Reviewer #1: Please carefully check the entire manuscript for a clear and unambiguous English language. Unfortunately, the presented result, discussion and interpretaion are not clear to me as a general reader. Please look overall for missing and correct words and ensure methodological clarity.

Reviewer #2: El estudio me pareció interesante, ya que permite observar un comportamiento estacional de la COVID-19, y el debilitamiento de la vigilancia de la Influenza en la región a consecuencia de la pandemia.

Reviewer #3: In general, the authors make use of very long sentences, difficult to understand. The text must be better redirected, the words sometimes seem unconnected, leaving the text unfluent and confusing.

Introduction - end of page 11 "and could be equal to or equal to RT-PCR" not understandable.

Methods - How was your sample defined? When was the data collection carried out? How were possible duplicate cases resolved? What was considered "clinical and epidemiological data"? I suggest making a diagram of the different types of information and their quantity (clinical cases, laboratory-confirmed cases, mortality).

Results - Were the two waves in the country in 2020 and 2021 between exactly the same weeks?

Discussion - Review the description of the acronyms. Absence of period in some sentences and others poorly placed.

6. PLOS authors have the option to publish the peer review history of their article (what does this mean?). If published, this will include your full peer review and any attached files.

Reviewer #1: No

Reviewer #2: **Yes: **Galo Guillermo Farfan Cano

Reviewer #3: No

---

## [Author Response · Author response to Decision Letter 0]

26 Jun 2023

Dear reviewers, thank you very much for your contributions to improving this manuscript, your opinions have been very valuable and correct.

---

## [Decision Letter · Decision Letter 1]

8 Oct 2023

PONE-D-23-01394R1The weekly P25 of the age of the influenza-like illness shows a higher correlation with COVID-19 mortality than rapid tests and could predict the evolution of COVID-19 pandemics in sentinel surveillance, Piura, Peru-2021.PLOS ONE

Dear Dr. OCANA GUTIERREZ,

Thank you for submitting your manuscript to PLOS ONE. After careful consideration, we feel that it has merit but does not fully meet PLOS ONE’s publication criteria as it currently stands. Therefore, we invite you to submit a revised version of the manuscript that addresses the points raised during the review process.

We look forward to receiving your revised manuscript.

Kind regards,

Jie Zhang

Academic Editor

PLOS ONE

Journal Requirements:

**Additional Editor Comments:**

Please revise the paper by addressing the reviewer's comments.

Reviewers' comments:

Reviewer's Responses to Questions

**Comments to the Author**

1. If the authors have adequately addressed your comments raised in a previous round of review and you feel that this manuscript is now acceptable for publication, you may indicate that here to bypass the “Comments to the Author” section, enter your conflict of interest statement in the “Confidential to Editor” section, and submit your "Accept" recommendation.

Reviewer #1: (No Response)

2. Is the manuscript technically sound, and do the data support the conclusions?

Reviewer #1: Partly

3. Has the statistical analysis been performed appropriately and rigorously? 

Reviewer #1: I Don't Know

4. Have the authors made all data underlying the findings in their manuscript fully available?

Reviewer #1: No

5. Is the manuscript presented in an intelligible fashion and written in standard English?

Reviewer #1: No

6. Review Comments to the Author

Reviewer #1: Although there were some efforts to improve the manuscript, unfortunatelly I still find the sentences difficult to understand. There are several spelling and grammatical errors. I would request to check the entire manuscript, specially for English language and clarity. Also not sure, if serological test (in addition to antigen tests) were used for laboratory diagnosis of SARS-CoV-2. Please confirm

7. PLOS authors have the option to publish the peer review history of their article (what does this mean?). If published, this will include your full peer review and any attached files.

Reviewer #1: No

---

## [Author Response · Author response to Decision Letter 1]

16 Nov 2023

Reviewer 1: I have incorporated all of your suggestions into my revision. They were helpful. Thank you.

---

## [Editor Report · Decision Letter 2]

21 Nov 2023

The weekly P25 of the age of the influenza-like illness shows a higher correlation with COVID-19 mortality than rapid tests and could predict the evolution of COVID-19 pandemics in sentinel surveillance, Piura, Peru, 2021.

PONE-D-23-01394R2

Dear Dr. OCANA GUTIERREZ,

We’re pleased to inform you that your manuscript has been judged scientifically suitable for publication and will be formally accepted for publication once it meets all outstanding technical requirements.

Kind regards,

Jie Zhang

Academic Editor

PLOS ONE

Additional Editor Comments (optional):

The authors have adequately addressed the reviewers' comments.
---

## [Editor Report · Acceptance letter]

11 Dec 2023

PONE-D-23-01394R2 

*The weekly P25 of the age of the influenza-like illness shows a higher correlation with COVID-19 mortality than rapid tests and could predict the evolution of COVID-19 pandemics in sentinel surveillance, Piura, Perú, 2021.*

Dear Dr. OCANA GUTIERREZ:

I'm pleased to inform you that your manuscript has been deemed suitable for publication in PLOS ONE. Congratulations! Your manuscript is now with our production department. 

Kind regards, 

on behalf of

Dr. Jie Zhang 

Academic Editor

PLOS ONE